**Data Availability Statement:** All relevant study data are provided in the Supporting Information (S1 Data) of the article.

# Trends in antimicrobial resistance amongst *Salmonella* Typhi in Bangladesh: A 24-year retrospective observational study (1999–2022)

Arif Mohammad Tanmoy[1,2], Yogesh Hooda[1], Mohammad Saiful Islam Sajib[1], Hafizur Rahman[1], Anik Sarkar[1], Dipu Das[1], Nazrul Islam[1], Naito Kanon[1], Md. Asadur Rahman[3], Denise O. Garrett[4], Hubert P. Endtz[2], Stephen P. Luby[5], Mohammod Shahidullah[6], Md. Ruhul Amin[7], Jahangir Alam[7], Mohammed Hanif[7], Samir K. Saha[1,8], Senjuti Saha[1] *

**1** Child Health Research Foundation, Dhaka, Bangladesh, **2** Department of Medical Microbiology and Infectious Diseases, Erasmus MC, Rotterdam, The Netherlands, **3** Popular Diagnostic Center, Dhaka, Bangladesh, **4** Sabin Vaccine Institute, Washington DC, Maryland, United States of America, **5** Division of Infectious Diseases and Geographic Medicine, Stanford University School of Medicine, Stanford, California, United States of America, **6** Department of Neonatology, Bangabandhu Sheikh Mujib Medical University, Dhaka, Bangladesh, **7** Department of Pediatrics, Bangladesh Institute of Child Health, Dhaka, Bangladesh, **8** Department of Microbiology, Bangladesh Shishu Hospital and Institute, Dhaka, Bangladesh

* senjutisaha@chrfbd.org

## Abstract

### Background

Rising antimicrobial resistance (AMR) in *Salmonella* Typhi restricts typhoid treatment options, heightening concerns for pan-oral drug-resistant outbreaks. However, lack of long-term temporal surveillance data on AMR in countries with high burden like Bangladesh is scarce. Our study explores the AMR trends of *Salmonella* Typhi isolates from Bangladesh, drawing comparisons with antibiotic consumption to optimize antibiotic stewardship strategies for the country.

### Methodology/Principal findings

The typhoid fever surveillance from 1999 to 2022 included two pediatric hospitals and three private clinics in Dhaka, Bangladesh. Blood cultures were performed at treating physicians' discretion; cases were confirmed by microbiological, serological, and biochemical tests. Antibiotic susceptibility was determined following CLSI guidelines. National antibiotic consumption data for cotrimoxazole, ciprofloxacin, and azithromycin was obtained from IQVIA-MIDAS database for comparison.

Over the 24 years of surveillance, we recorded 12,435 culture-confirmed typhoid cases and observed declining resistance to first-line drugs (amoxicillin, chloramphenicol, and cotrimoxazole); multidrug resistance (MDR) decreased from 38% in 1999 to 17% in 2022. Cotrimoxazole consumption dropped from 0.8 to 0.1 Daily defined doses (DDD)/1000/day (1999–2020). Ciprofloxacin non-susceptibility persisted at >90% with unchanged consumption (1.1–1.3 DDD/1000/day, 2002–2020). Low ceftriaxone resistance (<1%) was observed,

**Funding:** The work presented here was partially supported by Gavi, the Vaccine Alliance, through the World Health Organization-supported Invasive Bacterial Vaccine Preventable Diseases study (grant numbers 201588766, 201233523, 201022732, 200749550) to SKS, by leveraging the Pneumococcal Vaccines Accelerated Development and Introduction Plan (PneumoADIP) to SKS, and by the Bill and Melinda Gates Foundation through SEAP (grant number INV-008335) to DOG, SKS, and SS. The funding bodies had no role in the study design, data collection and analysis, decision to publish, or preparation of the manuscript.

**Competing interests:** The authors have declared that no competing interests exist.

with slightly rising MIC (0.03 to 0.12 mg/L, 1999–2019). Azithromycin consumption increased (0.1 to 3.8 DDD/1000/day, 1999–2020), but resistance remained ≤4%.

## Conclusion

Our study highlights declining MDR amongst *Salmonella* Typhi in Bangladesh; first-line antimicrobials could be reintroduced as empirical treatment options for typhoid fever if MDR rates further drops below 5%. The analysis also provides baseline data for monitoring the impact of future interventions like typhoid conjugate vaccines on typhoid burden and associated AMR.

## Author summary

Our study addresses the pressing issue of antimicrobial resistance (AMR) in *Salmonella* Typhi, which severely limits treatment options for typhoid fever. Globally, it also raises concerns for a potential pan-oral drug-resistant outbreak. We conducted typhoid fever surveillance, spanning 24 years (1999–2022) across two pediatric hospitals and three private clinics in Dhaka, the capital city of Bangladesh. Our aim was to understand how the AMR landscape of *Salmonella* Typhi has changed over the years and its correlation with antibiotic consumption. Our findings reveal a notable decline in resistance to first-line drugs including cotrimoxazole, whose decreasing consumption showed a statistically significant correlation with reduced resistance. Ciprofloxacin non-susceptibility persisted at >90%. Ceftriaxone resistance remained low, but there was an upward trend in its minimum inhibitory concentration (MIC). Azithromycin resistance is emerging and currently at <4% but is expected to increase due to the observed 38-fold rise in azithromycin consumption. Taken together, our results establish a baseline to assess the impact of interventions like typhoid conjugate vaccines on typhoid and associated AMR in Bangladesh. Our work also suggests that decreasing resistance to the first line of drugs in *Salmonella* Typhi may allow for the re-introduction of first line of antimicrobials as empirical treatment options for typhoid fever.

## Introduction

Antimicrobial resistance (AMR) poses a major threat to public health and was estimated to cause approximately 1.27 million deaths in 2019 [1]. If unchecked, deaths linked to AMR infections could multiply 10-fold by 2050 due to widespread antimicrobial use and a paucity of new antimicrobials [2,3]. In response to this threat, multiple strategies have been proposed, ranging from antibiotic stewardship to alternative antibiotic therapies. Implementing these interventions requires guidance from surveillance data, but many low- and middle-income countries (LMICs) suffer from large data deficits [1]. This study aims to provide insights into AMR trends in Bangladesh for *Salmonella enterica* serovar Typhi (*Salmonella* Typhi), the etiological agent of typhoid fever, over a 24-year period, from 1999 to 2022.

Typhoid fever is a pressing global public health concern, with the majority of cases occurring in south Asia [4]. In Bangladesh, the burden of typhoid is alarmingly high, with an estimated incidence of 913 per 100,000 person-years [5]. While the disease is more prevalent among school-age children [6], those under 2 years of age have the highest bacterial load

during infections by *Salmonella* Typhi [7]. Clinical and epidemiological data on typhoid fever have been traditionally reported from hospital-based surveillance studies, but, in many endemic countries, like Bangladesh, a large proportion of typhoid patients seek care at community-based diagnostic centers, private clinics, or hospital outpatient departments (OPD) [8–11]. This diverse care-seeking behavior complicates the precise estimation of the disease burden and AMR trends for typhoid fever.

Management of typhoid fever is becoming increasingly difficult with rising AMR. Multi-drug resistant (MDR, defined as the concurrent resistance to amoxicillin, chloramphenicol, and cotrimoxazole) *Salmonella* Typhi emerged in the 1970s [12,13]. This led to ciprofloxacin as the primary treatment choice by the mid-1980s, but widespread resistance followed. By the mid-2010s, over 90% of *Salmonella* Typhi in South Asia exhibited non-susceptibility to ciprofloxacin [14]. The current treatment options include azithromycin and third-generation cephalosporins such as ceftriaxone [13], but there are increasing reports of resistance to both the drugs [15–20]. The ongoing typhoid outbreak in Pakistan, caused by an extensively drug resistant (XDR; resistant to ceftriaxone, ciprofloxacin and first-line of antimicrobials) lineage [19], sparked further concerns. Additionally, a novel *acrB*-717 gene mutation linked to azithromycin resistance [16], has narrowed treatment options and raised fears of a pan-oral drug-resistant (PoDR) outbreak. A recent PoDR typhoid case has been reported from Pakistan and required intravenous carbapenem and colistin treatment [21]. This highlights the looming threat of AMR and related financial burden on patients and healthcare systems which are struggling to cope with high typhoid prevalence in LMICs [13,22].

To curb this high burden of typhoid fever, the Government of Bangladesh is considering the introduction of a typhoid conjugate vaccine (TCV). Phase-III trials have shown that TCV reduces typhoid fever by 81–85% and it has already been introduced in Pakistan, Liberia, Zimbabwe, Samoa, and Nepal [23,24]. However, the continuous monitoring of AMR trends in the post-vaccination era is essential to ensure sustained support from policymakers. This is especially crucial for Bangladesh, which is poised to shed its Least-Developed Countries (LDC) status in 2026 and start losing financial support from Gavi for vaccines. Amidst many competing priorities, policymakers of Bangladesh will require robust surveillance data to make informed decisions about the TCV program. To this end, our study provides insights on the historical trends in AMR through a comprehensive typhoid surveillance dataset of 12,435 *Salmonella* Typhi isolates, spanning 24 years (1999–2022). In this study, we aim to elucidate correlations between AMR and antibiotic consumption, providing a baseline to guide the design of antibiotic stewardship policies. It also provides insights into designing empirical treatment strategies for typhoid fever management and potential strategies to combat increasing AMR in Bangladesh.

## Methods

### Ethics statement

All study protocols received approval from the Ethics Review Committees (ERC) of Bangladesh Shishu Hospital and Institute (BSHI) at Dhaka, Bangladesh. Informed written consent and clinical information were obtained from parents/legal guardians for hospitalized cases. For out-patient cases, no formal consent was obtained as blood samples were collected as part of routine clinical care at the discretion of the treating physician and data from routine clinical care were retrospectively included without identifiable information.

### Enteric fever surveillance

The Child Health Research Foundation (CHRF) has been conducting enteric fever surveillance in Bangladesh since 1999, with a focus on the pediatric population (<18 years of age). This

surveillance is conducted at five sites in Dhaka, the capital of Bangladesh with a very high burden of typhoid fever [6]. These sites are: (a) Bangladesh Shishu Hospital and Institute (BSHI), a 650-bed hospital that is the largest pediatric hospital in the country, (b) Dr. M R Khan Shishu Hospital & Institute of Child Health (SSFH), a 250-bed hospital that is the second largest pediatric hospital in Dhaka, and (c) three branches of the Popular Diagnostic Center (PDC), a community-based consultation and diagnostic center. All cases in the hospitals were classified based on whether they were treated in the outpatient facility (OPD) or admitted to the hospitals (in-patient, IPD). All cases from PDC were classified as OPD. Blood samples were collected for microbiological culture as a part of the diagnostic service at the treating physicians' discretion. Both BSHI and SSFH were part of the Global Invasive Vaccine-Preventable Bacterial Disease Surveillance Network since 2009 [25]. Details of all three sites and our enteric fever surveillance program have been described previously [8,26]. All data used in this study are provided in Supporting Information, S1 Data.

## Etiological confirmation and antimicrobial susceptibility testing

In all hospitals, routine blood cultures were performed using standard methods, as described earlier [8,27]. In short, blood samples were sent to the microbiology laboratories of all the study hospitals where samples were cultured on blood, chocolate, and MacConkey agar plates. From 2016, blood specimens were incubated in BACTEC culture systems for up to 72 hours. BEEP-positive samples were then cultured on blood, chocolate, and MacConkey agar plates. Upon growth, biochemical (Klingler's Iron agar, Simmons citrate agar, motility-indole, and urea agar tests) and slide-agglutination tests (using *Salmonella* agglutinating antisera; Thermo Scientific, MA, USA) were done for identification of *Salmonella* Typhi. *Salmonella* Typhi isolates from PDC were reconfirmed at the CHRF laboratory using biochemical tests and slide agglutination tests. Antibiotic susceptibility tests were conducted for amoxicillin, cotrimoxazole, chloramphenicol, ciprofloxacin, ceftriaxone, and azithromycin using the Kirby-Bauer disc diffusion method (Oxoid, Thermo Scientific, MA, USA). The minimum inhibitory concentrations (MICs) for ceftriaxone and ciprofloxacin were determined for two subsets of isolates (from 1999–2019 and 1999–2016 respectively) using the broth microdilution method (Sigma-Aldrich, MO, USA) [28,29]. We determined MICs up to a maximum of 256 mg/L. All isolates collected before 2017 underwent MIC testing if available and recoverable on culture plates. Additionally, ceftriaxone MIC testing was performed on a random set of 299 isolates (considering hospital sites and year of isolation) collected between 2017 and 2019. Zone-diameters (ZD) and MIC data were interpreted based on the 2020 Clinical and Laboratory Standards Institute (CLSI) guidelines [30]. Azithromycin-resistant (ZD: <11 mm) and near-resistant (ZD: 11–13 mm) isolates were reconfirmed using Etest strips (bioMérieux, Marcy-l'Étoile, France). The $MIC_{50}$ for ceftriaxone and ciprofloxacin, indicating the MIC value at which 50% of tested isolates could not grow, was also calculated. Ceftriaxone MIC of two resistant isolates (MIC >256 mg/L) were excluded from all related analyses, as the exact MICs of those isolates could not be determined.

## Acquisition and analyses of antibiotic consumption data

To compare the patterns of typhoid cases, we acquired antibiotic consumption data (1999–2020) for cotrimoxazole, ciprofloxacin, and azithromycin from the IQVIA-Multinational Integrated Data Analysis System (IQVIA-MIDAS) database. This commercial database captures retail pharmacy sales, encompassing the total antibiotic volumes sold to both retailers and hospital pharmacies by wholesalers. It also records annual sales of each antimicrobial agent for each country. We acquired the antimicrobial sales data for the non-agricultural sector only.

Daily defined doses (DDD) for these antimicrobials were gathered from ATC/DDD Index 2023 (https://www.whocc.no/atc_ddd_index/). DDD represents the assumed average maintenance dose per day for its main indication in adults. Adjusting for population size, we calculated DDDs per 1000 inhabitants per day, following the WHO's Global Antimicrobial Resistance and Use Surveillance System (GLASS) [31]. Annual population data for Bangladesh was collected from the United Nations-World Population Prospect Report 2019 [32].

## Statistical analysis

We used Pearson's correlation coefficients to assess correlations between yearly AMR patterns and consumption of cotrimoxazole, ciprofloxacin, and azithromycin. All correlation tests were conducted in Stata v13.0. We also examined potential non-linear resistance patterns for ciprofloxacin and ceftriaxone using their MIC data, and for azithromycin with yearly resistance percentage data. The analyses for ciprofloxacin and ceftriaxone employed the generalized additive model (GAM) with the *mgcv* package in R (version R 4.2.3). The model formula was $y \sim s(x, bs = "cs")$, with 3,214 and 3,518 observations for ciprofloxacin and ceftriaxone, respectively. Yearly trend of azithromycin resistance was analyzed similarly in R (version R 4.2.3) using local polynomial regression.

## Results

### Resistance to the first-line antimicrobials—amoxicillin, chloramphenicol, cotrimoxazole

Our typhoid fever surveillance recorded 12,435 culture-confirmed *Salmonella* Typhi cases from 1999 to 2022 (S1 Data). Of these cases, 28% (3,478/12,435) were hospitalized (IPD) at either BSHI or SSFH (S1 Fig). During this period, we observed a declining trend in resistance among *Salmonella* Typhi isolates to the first-line antimicrobials—amoxicillin, chloramphenicol, and cotrimoxazole. In 2002, resistance peaked at 80%, but decreased to less than 20% in recent years (since 2020) for all three first-line antimicrobials (Fig 1A–1C). This decline was also reflected in multidrug resistance (MDR), which was 17% in 2022 (Fig 1D). The percentage of MDR isolates remained ≤26% since 2010 (average 20%; 18%–23% at 95% Confidence Interval, CI) and ≤19% since 2017 (average 17%; 15%–19% at 95% CI).

### Resistance to ciprofloxacin, ceftriaxone, and azithromycin

Non-susceptibility to ciprofloxacin in *Salmonella* Typhi was notably high, reaching 95% in 2001 from 56% in 1999, and remained above 90% throughout the study period. In 2022, 98% (1,266/1,298) of the isolates were non-susceptible to ciprofloxacin (Fig 2A). We further examined the non-susceptibility trend to ciprofloxacin using MIC data from 3,214 isolates recovered from 5,289 pre-2017 isolates (61%; Fig 3A). Our analysis with the Generalized Additive Model (GAM) of the MIC data revealed a consistent trend with no significant changes in non-susceptibility to ciprofloxacin. The MIC trend for ciprofloxacin remained stable at approximately 0.25 mg/L throughout the entire study period (Fig 3A). The yearly $MIC_{50}$ also remained unchanged since 2004 at 0.25 mg/L, except in 2012 ($MIC_{50}$ = 0.125 mg/L).

Ceftriaxone resistance, on the other hand, was very rare. Only two isolates were ceftriaxone-resistant, one isolate in 1999 and the other in 2000, with no further instances recorded (Fig 2B). To gain further insights, we generated ceftriaxone MIC data for 35% of the isolates (3,518/10,061) spanning from 1999 to 2019. Our analysis showed a gradual increase in MICs over time. Overall resistance to ceftriaxone increased from an MIC of 0.03 mg/L in 2001 to

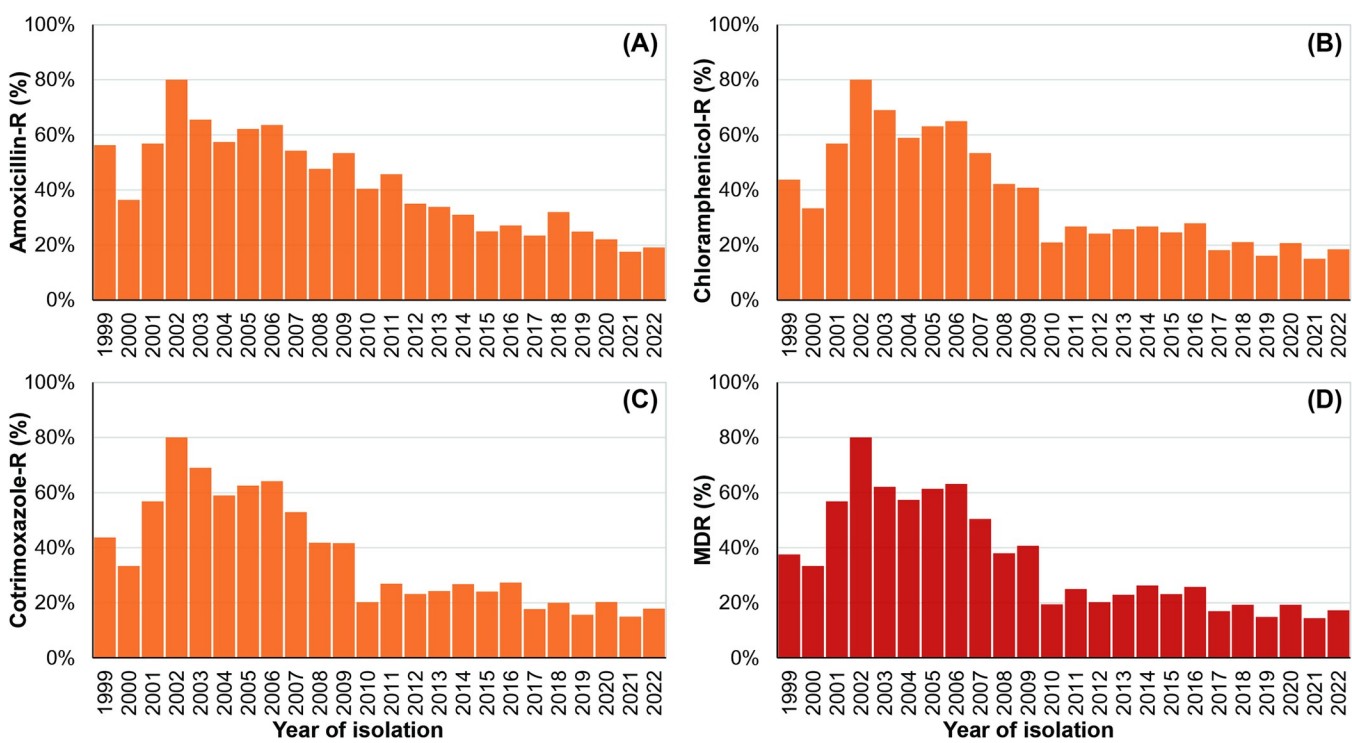

**Fig 1. Yearly trends in resistance (-R) and multidrug resistance (MDR) among *Salmonella* Typhi Cases (1999–2022, Bangladesh).** (A) Amoxicillin resistance, (B) Chloramphenicol resistance, (C) Cotrimoxazole resistance, and (D) MDR (defined as concurrent resistance to amoxicillin, chloramphenicol, and cotrimoxazole). Information on MDR is presented for all recorded cases.

0.12 mg/L in 2019 (Fig 3B). While these values remained below the breakpoint for reduced susceptibility (1.0 mg/L as per CLSI standards, Fig 3B), they represent a four-fold increase.

Azithromycin resistance in *Salmonella* Typhi was first identified in 2013 (Fig 2C). Subsequently, an average of 2% (1%–3%; 95% CI) of *Salmonella* Typhi isolates per year displayed resistance to azithromycin. An in-depth analysis using local polynomial regression to examine the yearly percentage of azithromycin-resistant isolates did not show any significant increase in azithromycin resistance since its initial identification in 2013.

## Correlation between AMR and antimicrobial consumption

We compared AMR data for cotrimoxazole, ciprofloxacin, and azithromycin with national antimicrobial consumption data (Fig 4). Cotrimoxazole consumption exhibited a notable decline, from 0.8 defined daily doses (DDD) per 1,000 persons per day in 1999 to 0.1 DDD/1,000 persons/day in 2020 (Fig 4A). Concurrently, resistance to cotrimoxazole showed a decreasing trend during this period (Figs 1C and 4A). Also, it showed significant correlation with consumption patterns (r = 0.77; p < 0.0005 at a 95% confidence interval, CI).

Ciprofloxacin consumption remained consistent throughout the study period, between 1.1 and 1.6 DDD/1,000 persons/day (Fig 4B). Non-susceptibility to ciprofloxacin showed a stable pattern as well, but no significant correlation with consumption was identified (r = 0.21; p = 0.38 at a 95% CI). On the other hand, azithromycin consumption increased by 38-fold between 1999 and 2020, from 0.1 DDD/1,000 persons/day to 3.8 DDD/1,000 persons/day. The average annual increase in azithromycin consumption was 0.18 DDD/1,000 persons/day (0.07–0.29 at a 95% CI) (Fig 4C).

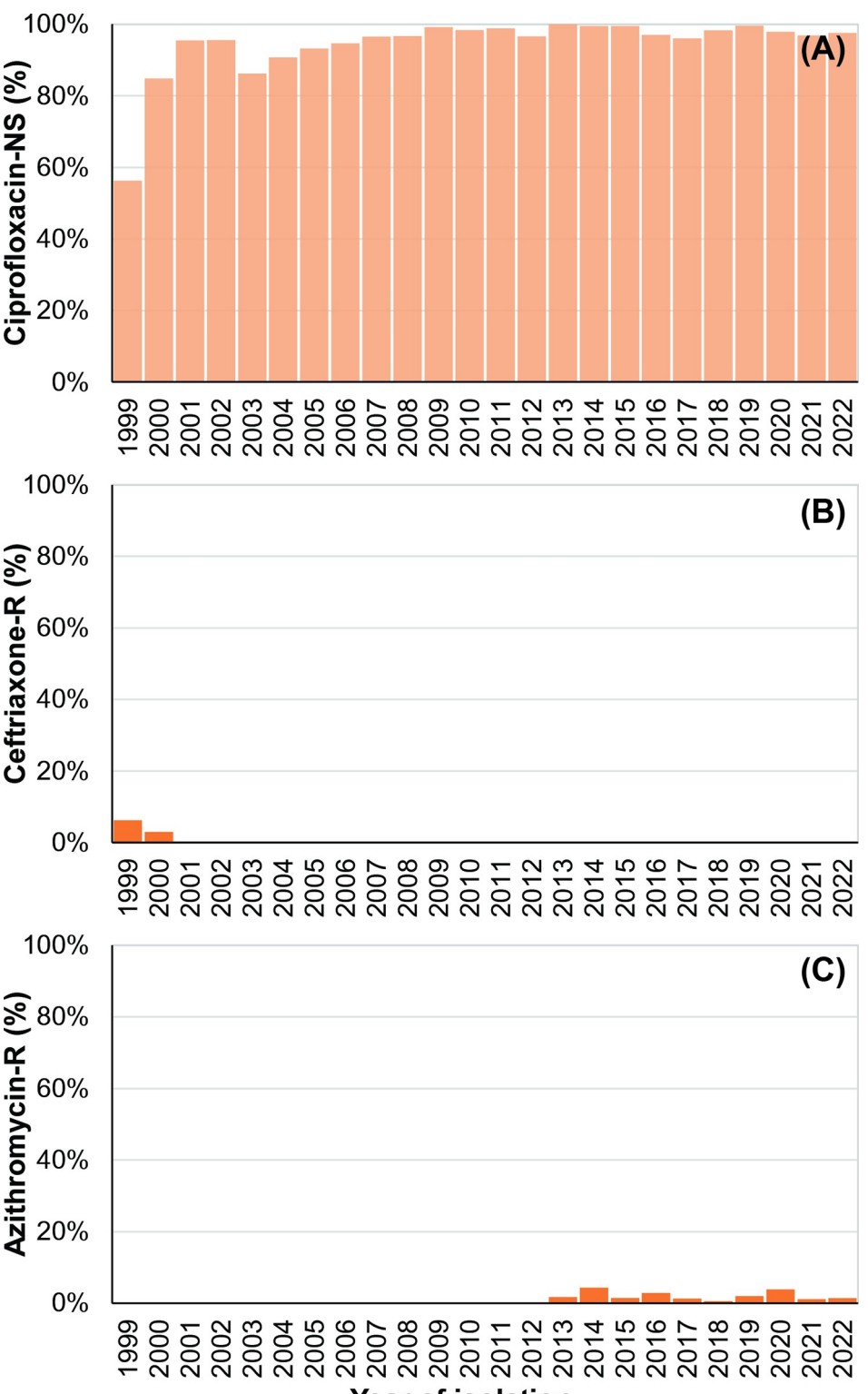

**Fig 2. Yearly trends in *Salmonella* Typhi susceptibility for Ciprofloxacin, Ceftriaxone, and Azithromycin (1999–2022) in Bangladesh.** Antimicrobial susceptibility data were present for 12,431, 12,414, and 8,187 isolates for (A) Ciprofloxacin non-susceptibility (-NS), (B) Ceftriaxone resistance (-R), and (C) Azithromycin resistance (-R), respectively.

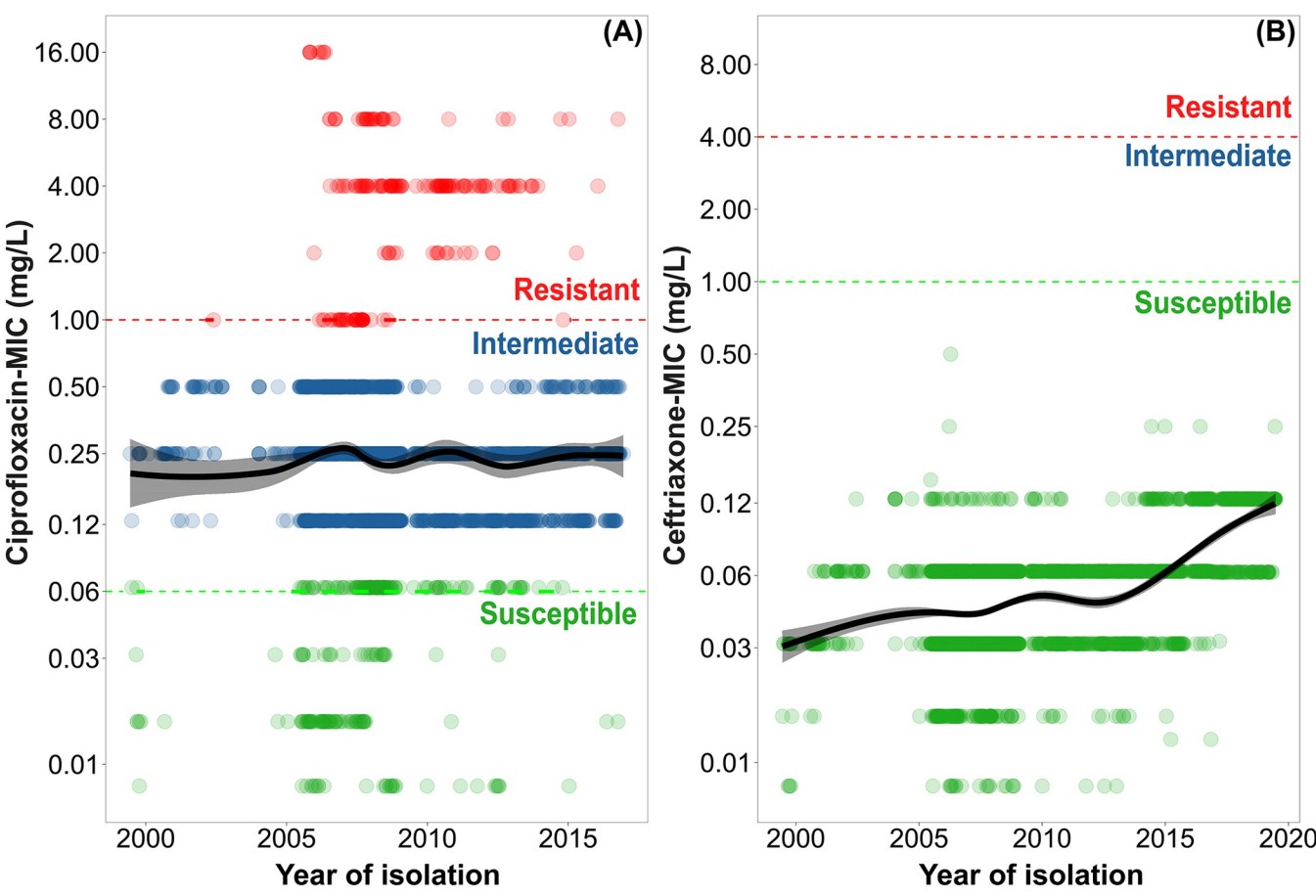

**Fig 3. Minimum inhibitory concentration (MIC) of ciprofloxacin and ceftriaxone among *Salmonella* Typhi cases in Bangladesh from 1999–2022.** The MICs were determined for a total of 3,214 and 3,518 isolates for (A) Ciprofloxacin (1999–2016), and (B) Ceftriaxone (1999–2019), respectively. The color of the dots represents resistant (red), intermediate (blue), and susceptible (green) isolates. The generalized additive model (GAM) lines depict the changes in the MICs (black line), with a 95% confidence interval (grey shades) over time. The y-axis is drawn based on a logarithm of 2 scale; MIC values are shown directly (for example, 0.5 instead of $2^{-1}$).

## Discussion

This study provides a comprehensive analysis of AMR patterns of 12,435 cases of typhoid fever in Bangladesh spanning 24 years. In resource-limited endemic countries like Bangladesh, a large proportion of typhoid fever cases is treated in the community, and more severe cases are treated in the hospital. This unique care-seeking behavior makes it challenging to estimate the true burden of typhoid fever and the patterns of AMR [10,11]. A previous study has shown that exclusively hospital-based surveillance might report a higher proportion of MDR typhoid cases compared to that in the community [10], not reflecting the real scenario. To bridge this gap, we established a typhoid surveillance system in Dhaka, Bangladesh, covering both hospital and community sites [5,8,33,34]. Popular Diagnostic Center (PDC), a community clinic and outpatient departments of the two study hospitals contributed 72% (8,957/12,435) of all the cases in our study.

Between 1999 and 2022, we observed a decline in resistance in *Salmonella* Typhi to the first-line antimicrobials- amoxicillin, chloramphenicol, and cotrimoxazole (Fig 1A-1C). This decline was associated with a decrease in multidrug resistance (MDR) (Fig 1D). Since 2017, over 80% of *Salmonella* Typhi isolates have remained susceptible to first-line drugs, with the

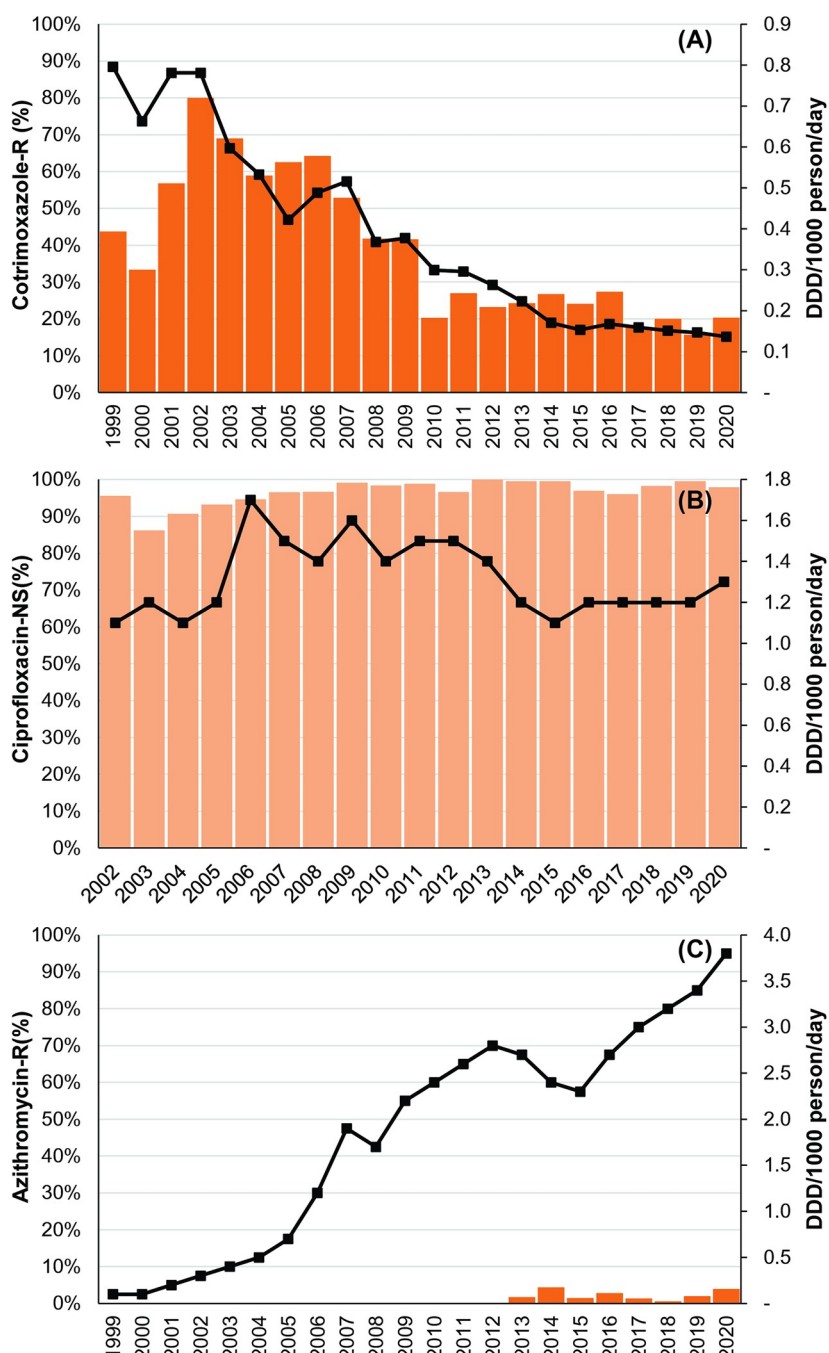

**Fig 4. Patterns of antimicrobial consumption and resistance patterns in *Salmonella* Typhi in Bangladesh from 1999–2020.** Patterns of antimicrobial consumption and resistance in (A) Cotrimoxazole, (B) Ciprofloxacin, and (C) Azithromycin among *Salmonella* Typhi cases in Bangladesh from 1999 to 2020. Consumption data for ciprofloxacin was unavailable from 1999 to 2001. Resistance (-R) or Non-susceptibility (NS) to cotrimoxazole, ciprofloxacin, and azithromycin are presented on the primary Y-axis. The secondary Y-axis displays antibiotic consumption, represented as the Defined Daily Dose (DDD) per 1,000 persons per day (DDD/1,000 day/year).

average MDR rate between 2017 and 2022 being 17% (15%–19%, 95% CI). Similar reductions in MDR rates have been documented in enteric fever surveillance studies across various countries, including India, Vietnam, Laos, Indonesia, Nepal, and Pakistan (pre-2016 XDR

outbreak) [35–42]. The Global Typhoid Genomics Consortium's report further supports decreasing MDR trends in several typhoid-endemic countries including Bangladesh, India, Nepal, Indonesia, and the Philippines [43]. We also noted a decline in cotrimoxazole consumption, a key first-line antimicrobial in Bangladesh (Fig 4A). It aligned with declining resistance to cotrimoxazole over time with a significant correlation (r = 0.77; p <0.0005 at 95% CI; Figs 1C and 4A). Further investigation is needed to understand the intricate relationship between drug consumption and resistance.

Both ciprofloxacin non-susceptibility and its consumption, however, remained high and unchanged throughout the study period, indicating its ongoing ineffectiveness for treating typhoid in Bangladesh. This stable resistance pattern over 24 years also suggests that changes in clinical outcomes due to ciprofloxacin use are unlikely in the near future. In addition, the lack of significant correlation between ciprofloxacin non-susceptibility and its use (r = 0.21; p = 0.38 at 95% CI) suggests the involvement of other factors, possibly genomic changes. Unlike MDR, ciprofloxacin non-susceptibility in *Salmonella* Typhi is generally driven by mutations in the *gyrA* and *parC* genes, which exert no fitness cost and are naturally maintained even if ciprofloxacin use is reduced [44].

The current empirical typhoid treatment in the country is either ceftriaxone or oral azithromycin. Only two ceftriaxone-resistant *Salmonella* Typhi isolates were identified in our study (in 1999 and 2000). However, analysis of 35% (3,518/10,061) of the isolates detected between 1999 and 2019, demonstrated a gradual four-fold increase in ceftriaxone MIC to 0.12 mg/L. Although still below the non-susceptibility breakpoint of 1.0 mg/L [30] and not clinically threatening yet, we must remain vigilant and continue to carefully monitor the gradual increase in ceftriaxone MIC. Thus, clinical consumption of this drug must also be carefully controlled, considering the narrowing window of treatment for typhoid.

Azithromycin resistance in *Salmonella* Typhi was first identified in our study in 2013 but remained sporadic with only 1% isolates exhibiting resistance in 2022 (Fig 2C). The consumption of the drug, on the other hand, increased by ~40-fold between 1999 and 2020. High azithromycin use may exert selection pressure, potentially fostering both macrolide- and non-macrolide resistance determinants in the gut [45]. The underlying molecular mechanism of azithromycin resistance in *Salmonella* Typhi is a single point mutation in the AcrB efflux pump gene, *acrB*-717, which has now been reported from multiple countries [16,43,46]. This mutation spontaneously emerges in diverse *Salmonella* Typhi genotypes [46]. Statistical modeling of a random *Salmonella* Typhi genome dataset from Bangladesh suggests a rising effective population size of azithromycin-resistant isolates [47]. However, our study does not demonstrate any increasing trend of azithromycin resistance. This sporadic resistance pattern could impact treatment guidelines, as azithromycin is a primary treating option for typhoid fever in Bangladesh. Spread of this resistance could limit its effectiveness and necessitate re-evaluating azithromycin use. Continuous monitoring is essential to ensure its efficacy and update guidelines to avoid compromising patient outcomes.

The introduction of the typhoid conjugate vaccine (TCV) in Bangladesh is expected to markedly reduce the burden of typhoid fever. The reduced number of cases in the post-TCV period will enable more focused efforts on implementing effective antimicrobial stewardship strategies in Bangladesh. A modeling study estimated that for each culture-confirmed case of typhoid, at least three additional patients receive antibiotics in Bangladesh [48]. TCV is also expected to significantly reduce the consumption of antibiotics in the country.

Before the global spread of MDR typhoid, first-line antimicrobials were widely used to treat the disease. These low-cost antimicrobials are well-tolerated and saved millions of lives for decades (1950s–1980s) [49–51]. To date, countries without MDR cases, like Samoa island, continue using amoxicillin as the primary treatment for typhoid [52,53]. In addition, successful

use of cotrimoxazole to treat typhoid has also been reported recently [54]. Since 1982, all three first-line drugs have been listed as essential medications by the Government of Bangladesh, which has helped prevent pharmaceutical price hikes [55]. Unlike the lengthy and expensive development and deployment of new antimicrobials [56], reintroducing these first-line drugs for treating typhoid cases requires no investment.

However, reintroducing older drugs also risks the re-emergence of MDR with increased use. The initial H58 *Salmonella* Typhi isolates in the 1990s carried MDR genes on an IncHI1 plasmid. Recent H58 *Salmonella* Typhi isolates in Bangladesh and other endemic countries have integrated these MDR genes into a chromosomal island. This integration helps avoid plasmid-imposed fitness costs, which negatively impact bacterial reproduction and survival due to plasmid carriage [47,57,58]. Due to these genomic changes, the re-emergence of MDR would require the frequency of H58 lineage to fall below 5% before reintroduction of older drugs. Continuous clinical, laboratory, and genomic surveillance in the post-TCV era will be essential to secure a safe window for the use of the old antimicrobials [43]. Monitoring antimicrobial use and resistance remains crucial for successfully treating typhoid and other infections effectively.

The results of our study should be considered within the context of several limitations. First, the two hospitals participating in the surveillance were among the largest pediatric reference hospitals in Bangladesh, resulting in a bias towards younger patients. Second, all study sites were in Dhaka, which may not represent other parts of Bangladesh. However, previous data reported 57% seropositivity for typhoid fever in Dhaka, reflecting the high number of cases in this city [6]. Third, many typhoid patients in Bangladesh are empirically treated without blood culture confirmation. No community-based enteric fever surveillance accounts for this, limiting comparisons. Fourth, the sampling strategy of our study did not include a calculated sample size, as it was conducted as part of an ongoing laboratory surveillance for enteric fever. Additionally, blood cultures were suggested by treating physicians, which may have introduced biases. Fifth, our broth microdilution plates were designed to determine MICs up to a maximum of 256 mg/L, and it prevented us from accurately measuring the exact ceftriaxone MICs of the two resistant isolates, which were excluded from subsequent analysis. Finally, national antibiotic consumption data from IQVIA-MIDAS database were used in this study, as regional or disease-specific data were unavailable. Since pharmaceutical sales data is proprietary information, we could not find any other source providing yearly comprehensive sales data of all antimicrobials available in Bangladesh. Consequently, we could not validate these data points with other sources. Additionally, antibiotic consumption data for amoxicillin, chloramphenicol, and ceftriaxone were unavailable.

Our study describes the trends in AMR in *Salmonella* Typhi isolates in Bangladesh over the last two decades, correlating them with antibiotic consumption. We propose a potential antibiotic stewardship strategy, reintroducing first-line antimicrobials for treating typhoid fever. Given that >80% of all *Salmonella* Typhi isolates have been susceptible to these drugs since 2017, this strategy could offer a viable empirical treatment option. Our analysis serves as a crucial baseline for monitoring the impact of future interventions, including the TCV and improvements in water, sanitation, and hygiene, on typhoid burden and AMR in the country. The next few years are crucial in the fight against typhoid fever. The baseline data presented here will play a key role in evaluating the effectiveness of interventions designed to reduce both the disease burden and antimicrobial resistance of this often-neglected tropical illness.

## Supporting information

**S1 Data. Antimicrobial resistance dataset of 12,435 typhoid cases in Bangladesh from 1999–2022 (S = Susceptible, I = Intermediate, and R = Resistant).** The dataset contains the

following information: ID–Anonymized ID of the isolates included in this study; PatientType–Department where the sample was collected (IPD: Hospital Inpatient Department, OPD: Hospital Outpatient Department, and Popular: Popular Diagnostic Center); Year–Year of blood collection; Amoxicillin–Resistance to amoxicillin using disk diffusion (S, I, and R); Chloramphenicol–Resistance to chloramphenicol using disk diffusion (S, I, and R); Cotrimoxazole–Resistance to cotrimoxazole using disk diffusion (S, I, and R); Ciprofloxacin–Resistance to ciprofloxacin using disk diffusion (S, I, and R); Ceftriaxone–Resistance to ceftriaxone using disk diffusion (S, I, and R); MDR–Multidrug resistance (MDR, and Not-MDR); Ciprofloxacin_MIC–MIC determined by broth microdilution for ciprofloxacin; Ceftriaxone_MIC–MIC determined by broth microdilution for ceftriaxone; Azithromycin–Resistance to azithromycin using disk diffusion and reconfirmed by Etest strips (S, I, and R).
(XLSX)

**S1 Fig. Overview of *Salmonella* Typhi database of 12,435 cases collected from 1999–2022 and MDR percentage in Bangladesh.** Yearly hospitalized (IPD) and outpatient (OPD) case numbers (on the left *y*-axis) are presented by the year of isolation. The *x*-axis labels indicate the number of isolates collected each year. Percentages of MDR for IPD and OPD cases per year represented as lines on the right *y*-axis.
(TIF)

## Acknowledgments

We would like to thank all past and present members of the Clinical Microbiology and Epidemiology teams of the Child Health Research Foundation (CHRF), Bangladesh for their routine data collection and discussions.

## Author Contributions

**Conceptualization:** Samir K. Saha, Senjuti Saha.

**Data curation:** Arif Mohammad Tanmoy, Mohammad Saiful Islam Sajib, Hafizur Rahman, Anik Sarkar, Dipu Das, Nazrul Islam, Md. Asadur Rahman, Senjuti Saha.

**Formal analysis:** Arif Mohammad Tanmoy, Yogesh Hooda, Mohammad Saiful Islam Sajib, Naito Kanon, Senjuti Saha.

**Funding acquisition:** Denise O. Garrett, Samir K. Saha, Senjuti Saha.

**Investigation:** Arif Mohammad Tanmoy, Mohammad Saiful Islam Sajib, Hafizur Rahman, Anik Sarkar, Dipu Das, Nazrul Islam, Samir K. Saha, Senjuti Saha.

**Methodology:** Arif Mohammad Tanmoy, Yogesh Hooda, Mohammad Saiful Islam Sajib, Hafizur Rahman, Anik Sarkar, Naito Kanon, Md. Asadur Rahman, Samir K. Saha, Senjuti Saha.

**Project administration:** Arif Mohammad Tanmoy, Denise O. Garrett, Mohammod Shahidullah, Md. Ruhul Amin, Jahangir Alam, Mohammed Hanif, Samir K. Saha, Senjuti Saha.

**Resources:** Hafizur Rahman, Naito Kanon, Denise O. Garrett, Hubert P. Endtz, Samir K. Saha, Senjuti Saha.

**Software:** Arif Mohammad Tanmoy, Yogesh Hooda, Mohammad Saiful Islam Sajib, Naito Kanon.

**Supervision:** Yogesh Hooda, Denise O. Garrett, Hubert P. Endtz, Samir K. Saha, Senjuti Saha.

**Validation:** Arif Mohammad Tanmoy, Mohammad Saiful Islam Sajib, Hafizur Rahman, Anik Sarkar, Dipu Das, Md. Asadur Rahman, Stephen P. Luby, Samir K. Saha, Senjuti Saha.

**Visualization:** Arif Mohammad Tanmoy, Yogesh Hooda, Mohammad Saiful Islam Sajib, Senjuti Saha.

**Writing – original draft:** Arif Mohammad Tanmoy, Yogesh Hooda, Senjuti Saha.

**Writing – review & editing:** Arif Mohammad Tanmoy, Yogesh Hooda, Denise O. Garrett, Hubert P. Endtz, Stephen P. Luby, Mohammod Shahidullah, Md. Ruhul Amin, Jahangir Alam, Mohammed Hanif, Samir K. Saha, Senjuti Saha.

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
