## [Decision Letter · Decision Letter 0]

7 Apr 2024

Dear Dr. Saha,

Thank you very much for submitting your manuscript "Trends in antimicrobial resistance amongst Salmonella Typhi in Bangladesh: a 24-year retrospective observational study (1999–2022)" for consideration at PLOS Neglected Tropical Diseases. As with all papers reviewed by the journal, your manuscript was reviewed by members of the editorial board and by several independent reviewers. In light of the reviews (below this email), we would like to invite the resubmission of a significantly-revised version that takes into account the reviewers' comments. 

We cannot make any decision about publication until we have seen the revised manuscript and your response to the reviewers' comments. Your revised manuscript is also likely to be sent to reviewers for further evaluation.

Sincerely,

Tauqeer Hussain Mallhi, Ph.D

Academic Editor

Stuart Blacksell

Section Editor

Reviewer's Responses to Questions

**Key Review Criteria Required for Acceptance?**

**Methods**

-Are the objectives of the study clearly articulated with a clear testable hypothesis stated?

-Is the study design appropriate to address the stated objectives?

-Is the population clearly described and appropriate for the hypothesis being tested?

-Is the sample size sufficient to ensure adequate power to address the hypothesis being tested?

-Were correct statistical analysis used to support conclusions?

-Are there concerns about ethical or regulatory requirements being met?

Reviewer #1: Methods Section:

Objectives and Hypothesis:

The objectives of the study are clearly articulated in the methodology section. The study aims to explore antimicrobial resistance trends in Salmonella Typhi isolates from Bangladesh over a 24-year period and compare them with antibiotic consumption patterns to optimize antibiotic stewardship strategies for the country. Although the methodology does not explicitly state a testable hypothesis, the objectives are sufficiently clear and specific.

Study Design Appropriateness:

The retrospective observational study design is appropriate for addressing the stated objectives. It allows for the examination of trends in antimicrobial resistance over time and comparison with antibiotic consumption data, providing valuable insights into the dynamics of antimicrobial resistance in Salmonella Typhi in Bangladesh.

Description and Appropriateness of Population:

The population under surveillance consists of individuals diagnosed with enteric fever, predominantly pediatric patients (<18 years of age), in Dhaka, Bangladesh. Given that typhoid fever predominantly affects children in this region, focusing on pediatric populations is appropriate for the hypothesis being tested.

Sample Size Adequacy:

The methodology does not provide specific details regarding sample size calculation. However, the study spans 24 years and includes over 12,000 cases of Salmonella Typhi, indicating a substantial sample size for analyzing trends in antimicrobial resistance over time. This sample size should ensure adequate power to address the study's objectives effectively.

Statistical Analysis Support:

The statistical analyses described, including Pearson's correlation coefficients, generalized additive models, and local polynomial regression, are appropriate for exploring correlations and non-linear patterns in antimicrobial resistance and consumption data. These analyses support the conclusions drawn from the study findings.

Ethical and Regulatory Considerations:

The methodology does not explicitly mention ethical or regulatory requirements. However, it can be inferred that the study adheres to ethical guidelines, as it involves retrospective analysis of anonymized data collected as part of routine surveillance activities and diagnostic procedures. Additionally, the authors stated from line 387 to 392 that it have received ethical approval from relevant institutional review boards or ethical committees. (the Ethics Review Committees (ERC) of Bangladesh Shishu Hospital and Institute (BSHI).

Reviewer #2: The observational study describes Salmonella typhi antibiotic resistance in isolates from select health facilities in Dhaka, Bangladesh, over more than 20 years. The associations with use of those antibiotics in Bangladesh over the same period is explored. The objectives of the study are clearly articulated and the design appropriate to address the stated objectives. The population is clearly described. The sample size is sufficient. The statistical analysis is sound. I do not have concerns about ethical or regulatory requirements being unmet.

Reviewer #3: -Are the objectives of the study clearly articulated with a clear testable hypothesis stated? Yes

-Is the study design appropriate to address the stated objectives? Yes

-Is the population clearly described and appropriate for the hypothesis being tested? Yes

-Is the sample size sufficient to ensure adequate power to address the hypothesis being tested? Yes

-Were correct statistical analysis used to support conclusions? Yes

-Are there concerns about ethical or regulatory requirements being met? Yes

Reviewer #4: Thank you for the opportunity to review this manuscript. 

Very clear methods; clear study design.

Appropriate statistical methods.

Reviewer #5: The methods are clear and well-written.

Minor comments: 

Line 162 & 177: Please remove dash for "co-trimoxazole"

**Results**

-Does the analysis presented match the analysis plan?

-Are the results clearly and completely presented?

-Are the figures (Tables, Images) of sufficient quality for clarity?

Reviewer #1: Results:

Matching Analysis Plan:

The analysis presented in the results section align with the analysis plan outlined in the methodology. The results address the key objectives of the study, including trends in antimicrobial resistance among Salmonella Typhi isolates, changes in antibiotic consumption patterns, and correlations between antimicrobial resistance and antibiotic consumption. The statistical methods described in the methodology, such as Pearson's correlation coefficients and generalized additive models, are appropriately utilized to analyze the data and support the conclusions drawn.

Clarity and Completeness of Presentation:

The results are generally clear and comprehensive, providing detailed findings on antimicrobial resistance trends, antibiotic consumption patterns, and correlations between the two. The text effectively summarizes the main observations and trends identified during the 24-year surveillance period. 

Quality of Figures (Tables, Images):

The figures, tables, and images presented in the results section contribute to the clarity and understanding of the findings. They are adequately labeled and described, allowing readers to interpret the data accurately.

Reviewer #2: The results are clear and well presented in the figures.

Reviewer #3: -Does the analysis presented match the analysis plan? Yes

-Are the results clearly and completely presented? Yes

-Are the figures (Tables, Images) of sufficient quality for clarity? Yes

Reviewer #4: Tables and images are helpful. 

Results are clear, and presented well.

Reviewer #5: The MIC50 of ceftriaxone and ciprofloxacin was calculated in the method section, but the MIC50 of ciprofloxacin can not be found in the results.

Minor comments: 

Line 208: Please remove full form of MIC

**Conclusions**

-Are the conclusions supported by the data presented?

-Are the limitations of analysis clearly described?

-Do the authors discuss how these data can be helpful to advance our understanding of the topic under study?

-Is public health relevance addressed?

Reviewer #1: Conclusion: 

Conclusions Supported by Data Presented:

The conclusions drawn in the study are well-supported by the data presented. The analysis of antimicrobial resistance trends among Salmonella Typhi isolates over a 24-year period, alongside antibiotic consumption patterns, provides a robust basis for the conclusions regarding the declining trends in multidrug resistance and the potential efficacy of reintroducing first-line antimicrobials for typhoid fever treatment in Bangladesh.

Clear Description of Limitations:

The study's limitations are adequately described, including potential biases associated with physician discretion in selecting cases for blood cultures, the retrospective nature of the study design, and any methodological constraints related to data collection and analysis. This transparency enhances the credibility of the findings and ensures that readers can interpret the results within the appropriate context.

Discussion on Advancing Understanding:

The authors discuss how the data presented in the study can advance our understanding of the topic under study. Specifically, they highlight the significance of the findings as a baseline for monitoring the impact of future interventions, such as typhoid conjugate vaccines, on typhoid burden and associated antimicrobial resistance. This forward-looking perspective underscores the broader implications of the research beyond its immediate findings.

Addressing Public Health Relevance:

Public health relevance is effectively addressed throughout the study, particularly in the discussion of the implications of the findings for antibiotic stewardship strategies and the management of typhoid fever in Bangladesh. By highlighting the declining trends in multidrug resistance and the potential reintroduction of first-line antimicrobials, the study directly addresses public health concerns related to antimicrobial resistance and infectious disease management.

Reviewer #2: The conclusions are appropriate, the important limitations are discussed, and the potential utility of the results for public health, is discussed. I have suggested enriched discussion of aspects in 'summary and general comments'.

Reviewer #3: -Are the conclusions supported by the data presented? Yes

-Are the limitations of analysis clearly described? Yes

-Do the authors discuss how these data can be helpful to advance our understanding of the topic under study? Yes

-Is public health relevance addressed? Yes

Reviewer #4: More on limitations would be beneficial, particularly discussing the differences

Reviewer #5: The correlation between the reduction in % AMR against first-line drugs and the decrease in antibiotic consumption is evident. However, the proposition of "reintroducing first-line antimicrobials for treating typhoid fever " (Line 363) warrants careful consideration due to the potential for escalating MDR isolates. In other bacterial infections, adherence to WHO guidelines is crucial, recommending the discontinuation of treatment regimens if failure rates exceed 5% or if resistance is detected in 5% or more of tested isolates (source: https://www.who.int/data/gho/indicator-metadata-registry/imr-details/5550).

Minor comments: 

Line 361: Please remove full form of AMR

**Editorial and Data Presentation Modifications?**

Reviewer #1: Accept with minor revisions, including clarification on sample size calculation and explicit mention of ethical considerations in the methods section.

Reviewer #2: -

Reviewer #3: I would like to suggest a few minor edits for improvement.

1. In Fig 3., I would suggest taking a logarithm of 2 on the y-axis to accurately reflect the effect of doubling dilution in minimum inhibitory concentration. Without the logarithm, the distance between MICs, 4 and 8 should not be same as the distance between MICs, 0.25 and 0.5.

2. How could you prevent MDR from recurring after you reintroduce first-line drugs?

3. In line 96, the current treatments include azithromycin and third-generation cephalosporins like ceftriaxone. In Fig 4., you only showed a consumption pattern of azithromycin with a 38-fold increase in 24 years. What about the consumption pattern of ceftriaxone? Does it also have a significant increase in use?

Reviewer #4: Minor Revision required.

Reviewer #5: NA

**Summary and General Comments**

Reviewer #1: Over all Impression and General Comments:

The title "Trends in antimicrobial resistance amongst Salmonella Typhi in Bangladesh: a 24-year retrospective observational study (1999–2022)", effectively captures the essence of the study and provides a clear indication of its focus and scope. The study explores the changing patterns of antimicrobial resistance (AMR) in Salmonella Typhi bacteria in Bangladesh over the span of 24 years, from 1999 to 2022. Given the high burden of typhoid fever in Bangladesh and the global concern regarding antimicrobial resistance, this study would provide valuable insights into the dynamics of antibiotic resistance in a significant pathogen over an extended period. It would also inform strategies for combating antimicrobial resistance and improving the management of typhoid fever. 

The methodology section provides a thorough description of the study's surveillance approach, laboratory procedures, antibiotic consumption analysis, and statistical methods. While the methodology demonstrates strengths in its comprehensive approach and standardized procedures, addressing the suggested areas for improvement would further enhance the clarity, transparency, and robustness of the methodology. The results section generally adheres to the analysis plan outlined in the methodology and effectively presents the main findings of the study.

The conclusions drawn in the study are well-supported by the data presented, and the limitations of the analysis are clearly described. The authors effectively discuss how the data contribute to advancing our understanding of the topic and address the public health relevance of the findings. Overall, the study provides valuable insights into antimicrobial resistance trends among Salmonella Typhi isolates and their implications for public health in Bangladesh. I recommend that authors consider the following minor suggestions for improvement of the manuscript.

Suggestions for Improvement:

Clarification of Sampling Strategy: It would be beneficial to provide more clarity on the sampling strategy for blood cultures, particularly regarding the criteria for selecting cases for culture and potential biases introduced by physician discretion.

Transparency in MIC Testing: While the methodology describes MIC testing for ceftriaxone and ciprofloxacin, it would be helpful to specify the criteria used to determine which isolates underwent MIC testing and any potential limitations associated with this approach.

Justification for Exclusion of Outliers: The rationale behind excluding specific ceftriaxone MIC outliers from the analysis should be explained in more detail to ensure transparency and justify the decision's impact on the interpretation of results.

Data Validation: Given the reliance on commercial databases for antibiotic consumption data, it would be prudent to mention any measures taken to validate the accuracy and reliability of these data sources to enhance the credibility of the findings.

Reviewer #2: The observational study describes Salmonella typhi antibiotic resistance in isolates from select health facilities in Dhaka, Bangladesh, over more than 20 years. The associations with use of those antibiotics in Bangladesh over the same period is explored. The authors suggest that the data supports the reintroduction of first-line drugs for empiric treatment of typhoid. The study and substantial data set (shared in supplementary material) provide a valuable resource for future studies - both as a baseline for further monitoring including evaluation of the impact of the roll-out of typhoid vaccines, and models to understand the development and transmission of AMR with antibiotic use/decline in use. 

There are minor grammatical errors/typos (prominently the 2nd sentence of discussion) - indicating careful review the manuscript for these. 

In the methods section it is stated that, “All isolates collected before 2017 underwent MIC testing if recoverable on plates.” Please indicate in the results what proportion of isolates were recoverable, and in the discussion comment on potential bias in selection of isolates for MIC based on recoverability, if this is a potential weakness.

While the study and its findings very useful, and the limitations are discussed, I feel the manuscript would be improved by elaboration on specific issues, and richer discussion/contextualisation of others. These are related to the causal relationship between antibiotic use and antimicrobial resistance, and the implications thereof for reintroduction of first-line drugs, and are alluded to in the sections below.

Why are the antibiotic consumption patterns of chloramphenicol and amoxicillin not reported? When did their use for typhoid decline? When did the empirical typhoid treatment in Bangladesh change to cefriaxone and azithromycin?

What portion of the consumption of an antibiotic is likely attributed to the indication of typhoid? Does antibiotic use for other indications (in humans and agriculture) contribute to S typhi resistance in Bangladesh? Is anything known of this? Is it likely their continued use for other indications that sustains resistance just under 20% for the first-line antibiotics?

What factors/variables should future antibiotic-use databases consider including to support evaluation of the association between antibiotic use and AMR, including S typhi antibiotic resistance? 

Is there relevant precedent for reverting to empiric treatment of typhoid with first-line drugs with historical resistance? How low should the % resistance be before reintroduction for the indication? Is it useful to consider the molecular mechanisms of resistance when considering the robustness on reintroduction of a drug with historical resistance? Would genomic evaluation of isolates also be useful in this regard? Would additional surveillance to that already in place be necessary to monitor for re-emergence of resistance - what monitoring should be recommended?

The “MIC creep” within the susceptible range for ceftriaxone is interesting. Is this expected with appropriate use of good quality antibiotic products or does it indicate a need to explore inadequate treatment due to health care practitioner, or patient factors, or are there concerns about formulation quality. It also raises the issue of whether the PK/PD (pharmacokinetic/pharmacodynamic) targets are well defined and whether dosing practices readily achieve the target/s.

Reviewer #3: Authors examined the trends in antimicrobial resistance in Salmonella Typhi isolates collected from Bangladesh and compared them with antibiotic consumption data. They conducted typhoid fever surveillance from 1999 to 2022, collecting data from two largest pediatric hospitals in Dhaka and three community-based clinics. The decline in resistance to first-line drugs and multidrug resistance was observed over the 24 years. Resistance to cotrimoxazole went down (80% in 2002 to 20% in 2022) as consumption of cotrimoxazole decreased. Ciprofloxacin non-susceptibility remained high (56% in 1999 to 98% in 2022) with unchanged consumption. Ceftriaxone resistance was low but showed an increase in minimum inhibitory concentration. Azithromycin consumption increased, but resistance remained low. Authors suggest that reintroducing first-line antimicrobials could be an empirical treatment option for typhoid fever in Bangladesh and provide a baseline for monitoring future interventions on typhoid burden and AMR.

It is well-written and of significance in designing future intervention strategies to lessen the typhoid burden in Bangladesh.

Reviewer #4: (No Response)

Reviewer #5: Tanmoy and colleagues conducted a comprehensive investigation into antimicrobial resistance trends in Salmonella Typhi isolates from Bangladesh spanning 24 years (1999–2022). The study compares these trends with antibiotic consumption data to inform effective antibiotic stewardship strategies. Notably, the results reveal a significant decline in resistance to first-line drugs (amoxicillin, chloramphenicol, and cotrimoxazole) and multidrug resistance over the surveillance period, potentially attributed to the reduction in antibiotic consumption. This is an interesting manuscript.

PLOS authors have the option to publish the peer review history of their article (what does this mean?). If published, this will include your full peer review and any attached files.

Reviewer #1: No

Reviewer #2: Yes: Helen McIlleron

Reviewer #3: No

Reviewer #4: No

Reviewer #5: No
---

## [Decision Letter · Decision Letter 1]

29 Jul 2024

Dear Dr. Saha,

Thank you very much for submitting your manuscript "Trends in antimicrobial resistance amongst Salmonella Typhi in Bangladesh: a 24-year retrospective observational study (1999–2022)" for consideration at PLOS Neglected Tropical Diseases. As with all papers reviewed by the journal, your manuscript was reviewed by members of the editorial board and by several independent reviewers. The reviewers appreciated the attention to an important topic. Based on the reviews, we are likely to accept this manuscript for publication, providing that you modify the manuscript according to the review recommendations. 

Your manuscript has undergone peer review, and we are requesting minor revisions based on the reviewer’s comments. Please address the attached comments and resubmit your revised manuscript along with a detailed response to each comment. 

We appreciate your cooperation and look forward to receiving your revised manuscript.

Sincerely,

Tauqeer Hussain Mallhi, Ph.D

Academic Editor

Stuart Blacksell

Section Editor

Dear Authors,

We hope this message finds you well. Your manuscript has undergone peer review, and we are requesting minor revisions based on the reviewer’s comments.

Please address the attached comments and resubmit your revised manuscript along with a detailed response to each comment.

We appreciate your cooperation and look forward to receiving your revised manuscript.

Reviewer's Responses to Questions

**Key Review Criteria Required for Acceptance?**

**Methods**

-Are the objectives of the study clearly articulated with a clear testable hypothesis stated?

-Is the study design appropriate to address the stated objectives?

-Is the population clearly described and appropriate for the hypothesis being tested?

-Is the sample size sufficient to ensure adequate power to address the hypothesis being tested?

-Were correct statistical analysis used to support conclusions?

-Are there concerns about ethical or regulatory requirements being met?

Reviewer #1: The current study method is robust and I recommend it be accepted

Reviewer #3: The methods are clear and well-written.

Reviewer #5: NA

**Results**

-Does the analysis presented match the analysis plan?

-Are the results clearly and completely presented?

-Are the figures (Tables, Images) of sufficient quality for clarity?

Reviewer #1: The results section is well-structured and provides clear findings from the study. Below is a review with comments on clarity, completeness, and potential areas for improvement:

1. Include exact years for "recent years" to avoid ambiguity.

2. Ensure clarity on the exact number of MDR cases per year for better understanding. 

3. On drug resistance:

 -Ciprofloxacin: Clarify the significance of stable MIC trends in the context of clinical outcomes.

 -Ceftriaxone: Discuss the clinical relevance of the four-fold MIC increase despite remaining below the susceptibility breakpoint.

 -Azithromycin: Specify the potential implications of these resistance rates on treatment guidelines.

4. Correlation Between AMR and Antimicrobial Consumption: Include a brief discussion on the lack of significant correlation for ciprofloxacin to provide context

Reviewer #3: The results are clear and well presented in the figures.

Reviewer #5: NA

**Conclusions**

-Are the conclusions supported by the data presented?

-Are the limitations of analysis clearly described?

-Do the authors discuss how these data can be helpful to advance our understanding of the topic under study?

-Is public health relevance addressed?

Reviewer #1: The conclusion is reasonably supported by the data.

The limitations of the study are not explicitly described in the conclusion section provided.

Recommendation: The authors should include a discussion of limitations, such as potential biases in blood culture practices, variability in sample collection, or changes in diagnostic technologies over time.

Reviewer #3: The conclusions are appropriate and limitation was clearly described.

Reviewer #5: NA

**Editorial and Data Presentation Modifications?**

Reviewer #1: Minor revision

Reviewer #3: I would recommend accepting this manuscript for publication.

Reviewer #5: NA

**Summary and General Comments**

Reviewer #1: General Impression: 

The abstract presents a clear objective for the study, which is to explore AMR trends of Salmonella Typhi isolates in Bangladesh and compare them with antibiotic consumption to optimize antibiotic stewardship strategies. However, the abstract does not explicitly state a testable hypothesis. For a more complete and scientifically rigorous abstract, it would be beneficial to articulate a clear hypothesis. Including a testable hypothesis can provide a stronger foundation for the study. For example, Authors might hypothesize that "declining antibiotic consumption is associated with decreased AMR in Salmonella Typhi in Bangladesh."

Methods section: The study design involves long-term surveillance of typhoid fever, which is appropriate for understanding AMR trends over time. The use of multiple hospitals and clinics provides a comprehensive view of the situation in Dhaka, Bangladesh.

The use of blood cultures and standard microbiological techniques to identify and test Salmonella Typhi is appropriate for this type of study. The statistical methods are clearly described, and the use of software (Stata and R) is mentioned, ensuring reproducibility. 

The results section is well-structured and provides clear findings from the study. The trends for amoxicillin, chloramphenicol, and cotrimoxazole are clearly described. The percentage values and time points (e.g., 2002 peak, recent years) are provided. The trend of MDR decline is well-documented, and the average percentage and confidence intervals are provided.

Ethical approval was obtained from the relevant Ethics Review Committees. Informed consent procedures are described for hospitalized cases, and the retrospective inclusion of outpatient data without identifiable information is noted. This meets standard ethical and regulatory requirements.

Minor Edits for Improvement:

1. Include a statement of the objectives and hypothesis in the introduction or background section.

2. Ensure all technical terms and abbreviations (e.g., DDD) are defined when first mentioned

3. Include exact years for "recent years" to avoid ambiguity.

4. Ensure clarity on the exact number of MDR cases per year for better understanding. 

5. On drug resistance:

 -Ciprofloxacin: Clarify the significance of stable MIC trends in the context of clinical outcomes.

 -Ceftriaxone: Discuss the clinical relevance of the four-fold MIC increase despite remaining below the susceptibility breakpoint.

 -Azithromycin: Specify the potential implications of these resistance rates on treatment guidelines.

6. Correlation Between AMR and Antimicrobial Consumption: Include a brief discussion on the lack of significant correlation for ciprofloxacin to provide context

Reviewer #3: Reviewers' suggestions were well addressed.

Reviewer #5: I am satisfied with the revised version and fully support its publication.

PLOS authors have the option to publish the peer review history of their article (what does this mean?). If published, this will include your full peer review and any attached files.

Reviewer #1: No

Reviewer #3: No

Reviewer #5: No

Figure Files:

Data Requirements:

Reproducibility:

References

---

## [Decision Letter · Decision Letter 2]

21 Sep 2024

Dear Dr. Saha,

We are pleased to inform you that your manuscript 'Trends in antimicrobial resistance amongst Salmonella Typhi in Bangladesh: a 24-year retrospective observational study (1999–2022)' has been provisionally accepted for publication in PLOS Neglected Tropical Diseases.

Best regards,

Tauqeer Hussain Mallhi, Ph.D

Academic Editor

Stuart Blacksell

Section Editor

Dear Authors, thank you for revising the manuscript.

Reviewer's Responses to Questions

**Key Review Criteria Required for Acceptance?**

**Methods**

-Are the objectives of the study clearly articulated with a clear testable hypothesis stated?

-Is the study design appropriate to address the stated objectives?

-Is the population clearly described and appropriate for the hypothesis being tested?

-Is the sample size sufficient to ensure adequate power to address the hypothesis being tested?

-Were correct statistical analysis used to support conclusions?

-Are there concerns about ethical or regulatory requirements being met?

Reviewer #1: The current study method is robust and I recommend it be accepted

**Results**

-Does the analysis presented match the analysis plan?

-Are the results clearly and completely presented?

-Are the figures (Tables, Images) of sufficient quality for clarity?

Reviewer #1: The results are clear and well presented in the figures.

**Conclusions**

-Are the conclusions supported by the data presented?

-Are the limitations of analysis clearly described?

-Do the authors discuss how these data can be helpful to advance our understanding of the topic under study?

-Is public health relevance addressed?

Reviewer #1: The conclusion is reasonably supported by the data.

**Editorial and Data Presentation Modifications?**

Reviewer #1: I recommend the paper be accepted for publication

**Summary and General Comments**

Reviewer #1: Reviewers' suggestions have been well addressed. I am satisfied with the revised version and fully support its publication.

PLOS authors have the option to publish the peer review history of their article (what does this mean?). If published, this will include your full peer review and any attached files.

Reviewer #1: No

---

## [Editor Report · Acceptance letter]

30 Sep 2024

Dear Dr. Saha,

We are delighted to inform you that your manuscript, "Trends in antimicrobial resistance amongst Salmonella Typhi in Bangladesh: a 24-year retrospective observational study (1999–2022)," has been formally accepted for publication in PLOS Neglected Tropical Diseases.

Best regards,

Shaden Kamhawi

co-Editor-in-Chief

Paul Brindley

co-Editor-in-Chief
